# A simple and low-cost electrode based on Nafion-stabilized silver nanoparticles supported on FTO for the electrochemical determination of Pb (II) and Cu (II)

**Leonardo J. Monroy-Cruz**[1], **Akemi Morales-Kato**[1], **Yndira Dolores-Maldonado**[1], **Katiuska Castro**[1], **Alen Zimic-Sheen**[1], **María Belén Balta**[1], **Geraldine J. Otayza-Melgarejo**[1], **Raúl León**[2], **Patricia Sheen**[1], **Wilner Valenzuela** (ORCID)[1,3]*, **Mirko Zimic** (ORCID)[1]*

**1** Laboratorio de Bioinformática, Biología Molecular y Desarrollos Tecnológicos. Laboratorios de Investigación y Desarrollo. Facultad de Ciencias e Ingeniería. Universidad Peruana Cayetano Heredia, Lima, Perú, **2** Laboratorio de Metalurgia y Ciencias de Materiales, NDT Innovations, Inc., Lima, Perú, **3** Grupo de Investigación Electroquímica Aplicada. Facultad de Ciencias. Universidad Nacional de Ingeniería, Lima, Perú

* mirko.zimic@upch.pe (MZ); wvalenzuelab@uni.pe (WV)

## Abstract

Increasing awareness of the environmental risks posed by heavy metal accumulation in the environment—due to their toxicity and persistence in biological systems—has driven the development of more efficient and accessible detection methods. Conventional techniques, despite their accuracy, are often expensive, time-consuming, and reliant on non-portable specialized equipment. This study presents a novel, low-cost electrochemical sensor using a fluorine-doped tin oxide (FTO) electrode modified with Nafion-stabilized silver nanoparticles (AgNPs) for the rapid and accurate detection of Pb (II) and Cu (II) in water samples. The electrode preparation involved the ultrasonic cleaning of the FTO, followed by its surface modification with Nafion and the electrodeposition of AgNPs. Electrochemical and structural characterization confirmed the advantages of this approach, showing a significant improvement in conductivity and in the active surface area of the electrode, which allowed for the sensitive detection of the target metals. The optimization of analytical parameters, including accumulation time, deposition potential, and pH, facilitated the effective determination of the analytes by differential pulse anodic stripping voltammetry (DPV). The results demonstrated low detection limits of 8.87 ppb for Pb (II) and 3.26 ppb for Cu (II), suitable for in-situ applications in environmental monitoring according to environmental quality standards. The sensor's portability, coupled with its low cost and rapid analysis capability, addresses critical challenges in current monitoring practices and opens new avenues for widespread environmental surveillance in remote areas such as the Andean regions, where heavy metal contamination is a significant concern.

## Introduction

The dynamics of interaction between biotic and abiotic components within the environment play a crucial role in maintaining a balance suitable for human development. This balance

**Data availability statement:** All relevant data are within the paper and its Supporting Information files.

**Funding:** The author(s) received no specific funding for this work.

**Competing interests:** The authors have declared that no competing interests exist.

derives from a synergy among the integration of its physical, chemical, biological, and meteorological properties, creating spaces conducive to life. The harmonious interaction of natural and anthropogenic factors is vital for the dynamics of these spaces, underscoring the importance of sustainable practices and proper management of natural elements along with human interventions to ensure ecological sustainability and the prosperity of populations [1].

In this context, heavy metals, although natural components of the environment, become a threat when their release is exacerbated by human activities such as mining, agriculture, and refining. Particularly in Peru, the situation is alarming, with around 6128 mining environmental liabilities identified as primary sources of heavy metal pollution recorded by the end of 2023 [2], highlighting the negative impact of inefficient waste management systems. This problem represents not only an environmental challenge but also an urgent public health issue, especially considering the harmful effects of exposure to heavy metals from an early age, including irreversible neurological damage, hepatotoxicity, gastrointestinal and circulatory issues, skin inflammation and cancer [3,4].

Specific geographic areas in the Andean region of Peru, such as Huancavelica, Cerro de Pasco, and the Callejón de Huaylas (Ancash), face critical levels of mercury (Hg), lead (Pb), and arsenic (As) pollution, respectively [5–8]. These pollutants, when accumulated in the body from early ages, can cause profound and persistent neurological damage, underscoring the need for effective and accessible interventions. However, rural Andean communities, being the most affected by this issue and often with limited resources, face significant barriers to accessing treatments such as chelation, which is essential for removing these metals from the body [9,10].

Some of the most studied heavy metals due to their dispersion, as well as their toxicity, are Hg, Cu, Cd, Pb, Cr, As, among others [11]. Lead, for example, can damage the central nervous system, kidneys, and blood system [12,13]. Similarly, copper, despite being an important ion for biochemical processes in humans, can cause problems in the hepatic and gastrointestinal systems when present in high concentrations [14]. The acceptable blood concentration of Pb is below 5 μg/dL for adults and 3.5 μg/dL for children. Levels exceeding 70 μg/dL in adults and 20 μg/dL in children are considered critical [15]. For copper (Cu), permissible concentrations range from 63.5 to 158.9 μg/dL, while higher levels are considered hazardous to health [16].

The development of innovative methods for the detection and quantification of heavy metals is fundamental to address the environmental and public health challenges faced by Andean countries, especially in regions like Peru, where pollution from heavy metals due to anthropogenic activities has had a significant impact. The ability to perform accurate detections near affected communities provides a critical opportunity to alert health authorities to the presence of dangerous levels of metals, facilitating early assessment and treatment of children and other vulnerable individuals exposed to high concentrations of these toxins [17–19].

Currently, methods routinely employed for the detection of heavy metals in environmental samples include atomic absorption spectrometry (AAS) [20], cold vapor technique [21], inductively coupled plasma-optical emission spectrometry (ICP–OES) [22] and inductively coupled plasma mass spectrometry (ICP-MS) [23,24]. However, these methods present disadvantages such as extensive analysis time and high cost. Additionally, they require a high specialization in the use of these equipments and are not portable, being this latter a problem when environmental *in situ* analyses are required [25,26]. Therefore, there is an urgent need for economical and reliable field methods for metal detection. Electrochemical techniques have emerged as a new alternative for analysis because, just like conventional techniques, they present low limits of detection and quantification, high reproducibility, and good sensitivity in a shorter analysis time. Moreover, they enable simultaneous multi-elemental analysis of an electrolytic solution, since some of these techniques differentiate between the oxidation

states of various analytes allowing for greater selectivity and specificity, which is fundamental when studying certain ions [27,28]. For example, Cr in a +3 state is essential for some cellular processes, however, in a +6 state, it is carcinogenic [29].

One of the electrochemical techniques used for the detection of heavy metals is differential pulse anodic stripping voltammetry (DPV). This technique consists of two main stages: preconcentration and dissolution [30]. In the first part of the analysis, voltages are applied to the system that allows for the reduction of the analytes present, thereby pre-concentrating the metals on the electrode surface. Subsequently, the metals are oxidized, causing them to dissolve in solution. This last process generates characteristic signals for each metal according to its REDOX potential [31]. The pre-concentration stage of anodic stripping voltammetry allows for greater sensitivity than other techniques. Furthermore, voltage arrangements such as differential pulse or square wave differential pulse allow for even lower detection limits because, with the help of these voltage arrangements, it is possible to eliminate the signal corresponding to non-faradaic current, ensuring that the current signals correspond only to the REDOX processes on the material's surface [32].

Fluorine-doped tin oxide (FTO) is an alloy that presents a tetragonal structure and has been employed in various areas such as sensors, digital displays, electrode substrates, among others. It exhibits high chemical and thermal stability, high conductivity, low toxicity, and its manufacturing is low cost [33,34]. Due to these properties, FTO is used in the development of electrochemical detection methods. However, systems based on FTO in some cases do not achieve sufficiently high signal intensities for certain heavy metals, so its use does not reach to detect certain metals at low concentrations. Therefore, structural modifications are made to this material to increase the stability of FTO and its affinity with the analyte [35–37]. For this purpose Nafion is often used, which is a polymer that possesses a negative charge and allows increasing the affinity for cationic ions [38]. FTO-based electrodes can be modified with some conductive material that improves the response and sensitivity. One way to achieve this is by adding metallic nanoparticles. The properties of materials at the nanoscale provide valuable properties in electroanalysis such as improved diffusion, a high active surface area that enhances the electrocatalytic activity of the electrode, or particular optical properties. Silver nanoparticles (AgNPs) cause a significant increase in the signal and are widely used due to their good electrical conductivity and ease in the synthesis method [24,39–41].

In the present study, an FTO electrode modified with silver nanoparticles stabilized with Nafion (AgNPs/Nf/FTO) was developed for the detection of heavy metals in water samples using the DPV technique.

## Materials and methods

### Materials and reagents

All chemical reagents used in the experiments are of analytical reagent grade and were used directly without further purification. Disodium hydrogen phosphate ($Na_2HPO_4$, ≥ 98.5%), monopotassium phosphate (≥99% purity), perfluorinated resin solution of Nafion (20% w/w), copper (II) chloride dihydrate (35.0–38.0% Cu), basic lead acetate (72–75% Pb), ammonium hydroxide (28.0–30.0% $NH_3$, ≥99.99% purity), and FTO-coated glass (plates of 50 mm x 50 mm x 2.2 mm, surface resistivity ~7 Ω/sq) were acquired from Sigma-Aldrich (United States). Silver nitrate (99.4% purity) was purchased from Fisher Scientific Company, and cadmium chloride (99.4% purity) was obtained from J.T. Baker Chemical Company Co. A phosphate buffer solution (0.1 M, pH 3.5, PBS) was prepared by mixing 0.2366 g of disodium hydrogen phosphate ($Na_2HPO_4$, ≥ 98.5%) and 0.9994 g of monopotassium phosphate ($KH_2PO_4$, ≥ 98%) on an analytical balance and then adjusting the pH with 0.1 M solutions of NaOH and HCl. The

electrochemical assays were conducted with a PalmSens4 potentiostat (PalmSens B.V., Netherlands), using a three-electrode system. The FTO conductive glass was precisely cut using a diamond-tipped disc technique into fragments of 20 mm × 10 mm × 2.2 mm, which were used as working electrodes. A platinum wire served as the counter electrode, and an Ag/AgCl electrode was used as the reference electrode, containing a 3 M KCl solution.

### Preparation of AgNPs/Nf/FTO electrode

The FTO glass was cleaned ultrasonically in a 2 mL Eppendorf tube with ethanol for 15 minutes in a sonicator; after the process, it was rinsed. The same process was repeated with ultrapure water. After cleaning the electrode, the working area of the FTO was delimited to 0.5 cm². Then, the electrode surface was modified with Nafion (Nf/FTO) by immersing the FTO in a 5% w/w Nafion solution for 20 seconds; subsequently, it was left to dry for 30 minutes at room temperature. Finally, silver nanoparticles (AgNPs) were electrodeposited on the Nf/FTO using the cyclic voltammetry (CV) technique at a sweep rate of 0.05 V/s within a voltage range of -0.3 to 0.6 V, following the protocol of Nia et al. (2015) [24].

### Electrochemical characterization

To understand the REDOX behavior and the conductivity of the support material at each stage of electrode modification, CV was employed. For this purpose, a PalmSens4 potentiostat with the previously mentioned three-electrode system was used. The potential range used was -0.75 to 0.6 V, with a sweep rate of 0.05 V/s and an equilibration time of 10 s. Under these parameters, the FTO, Nf/FTO, and AgNPs/Nf/FTO electrodes were characterized using a 0.1 M PBS buffer solution at pH 3.5.

### Characterization of AgNPs/Nf/FTO electrode

Morphological analyses of the electrode surface at different stages of the modification process (unmodified FTO-glass, Nf/FTO, AgNPs/Nf/FTO) were carried out using a scanning electron microscope (SEM), specifically the JEOL JSM 6010LV (Tokio, Japan), operated at a magnification of 10,000 X and a discharge voltage of 20 kV. To optimize surface visualization, the electrodes were coated with a thin gold layer of approximately 15 nm thickness, which were deposited by sputtering [42,43].

### Determination of analytes by differential pulse anodic stripping voltammetry (DPV)

Metal ions were analyzed by DPV in a PBS buffer solution. After a 10 second equilibration period, a DPV potential scan was conducted, covering a range from -0.75 V to 0.1 V, with a sweep rate of 0.05 V/s and a step potential of 0.01 V. Following the analysis, a voltage of 0.1 V was applied for 60 seconds to clean the electrode. For the determination of analytes, both the deposition time and voltage, along with pH evaluation, were optimized to establish the optimal conditions for analysis according to previous protocols [27,35]. Additionally, to verify an improvement in the sensor's performance after the modifications, a DPV analysis of standard Cu and Pb solutions was conducted with electrodes at each modification stage (unmodified FTO-glass, Nf/FTO, AgNPs/FTO, and AgNPs/Nf/FTO).

### Determination of analytes in real samples

The determination of Pb (II) and Cu (II) in real acid mine drainage samples collected in the Ancash department, Peru, was performed. For the analysis, a 5 mL aliquot of the sample was

taken to which 45 mL of PBS buffer was added. The analysis by DPV used the same parameters employed in the experiments previously conducted in the present study with standard solutions. To compare the results of our sensor, an analysis was conducted to determine the concentration of Pb (II) and Cu (II) using the ICP-MS method. For this purpose, the services of SGS, a company accredited to perform this procedure according to the guidelines established by the United States Environmental Protection Agency [44], were employed.

## Results

### Electrochemical and surface characterization of the AgNPs/Nf/FTO electrode

The cyclic voltammetry curves from the different functionalization stages showed changes in current intensity and differences in potential locations. The unmodified FTO-glass electrode stage shows a reduction peak at a voltage of -0.3 V, obtaining current intensities between -110 and 50 µA (Fig 1). The second stage of functionalization (Nf/FTO) shows a decrease in current intensity across the entire voltage window compared to the voltammogram obtained in the analysis of the unfunctionalized FTO electrode. On the other hand, the AgNPs/Nf/FTO electrode showed an oxidation peak at a voltage of 0.2 V. Likewise, a significant increase in current was observed compared to the other modification stages (Fig 1).

Scanning Electron Microscopy (SEM) characterization managed to show the surface of the unmodified FTO electrode (Fig 2A). Thereafter, an electrode modified only with silver nanoparticles (AgNPs/FTO) was visualized, in which globular aggregations corresponding to the nanoparticles were observed (Fig 2B). In Fig 2C, the morphology of the AgNPs/Nf/FTO electrode is observed, where the nanoparticle clusters are also visible. Additionally, the nanoparticle areas were measured using the open-source image processing software ImageJ2 (version 2.14.0/1.54f), revealing an average size range of 35–42 nm, this is consistent with previous studies that employ the same nanoparticle deposition protocol [24]. Finally, Fig 1D corresponds to an electrode on which a cleaning protocol was applied for its recycling in

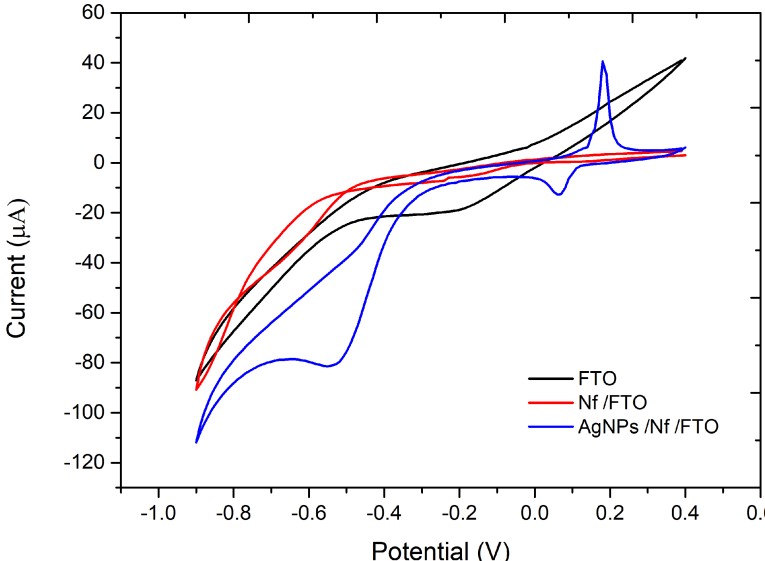

**Fig 1. Cyclic voltammograms of the different functionalization stages of the working electrode.** Unmodified FTO-glass (black), Nf/FTO (red), AgNPs/Nf/FTO (blue) in 0.1 M PBS at a sweep rate of 0.05 V/s.

another determination. In this figure, the surface has the same characteristics compared to the unmodified FTO-glass (Fig 2A).

## Influence of accumulation time

The influence of deposition time on the peak current was studied over a range of 30–150 seconds. Experiments were conducted in 0.1 M PBS buffer solutions with a pH of 3.5 containing 1 ppm of Pb (II) and Cu (II). The results are shown in Fig 3A and 3B. For both analytes, it was observed that the current signal increased with deposition time up to 90 seconds. After this point, a constant decrease in the signals of both metals was observed.

## Effect of deposition potential

The effect of the preconcentration potential on the anodic dissolution peak was studied in the range of -0.5 to -0.8 V and -0.5 to -0.9 V for Pb (II) and Cu (II), respectively. Preconcentration was carried out over 90 seconds in 0.1 M PBS buffer solutions with a pH of 3.5 containing 1 ppm of Pb (II) and Cu (II). The results are shown in Fig 3C and 3D. It is observed that the peak intensity for Pb (II) and Cu (II) ions reached their maximum at reduction potentials of -0.7 V and -0.8 V, respectively; thereafter, the peak current decreases. Also, at more negative potentials, wear of the support material was observed, leading to a significant decrease in current intensity.

## Effect of pH

The analysis of the influence of pH on the signal obtained in the determination of the analytes was carried out using 6 points in the pH range of 2.5 to 5 based on information extracted from the Pourbaix diagrams for Pb (II) and Cu (II) ions [45]. Also, a value of -0.7 V was used for the deposition voltage with 90 seconds of accumulation time. The results from Fig 3E and 3F show an increase in the signal magnitude for both elements towards more acidic

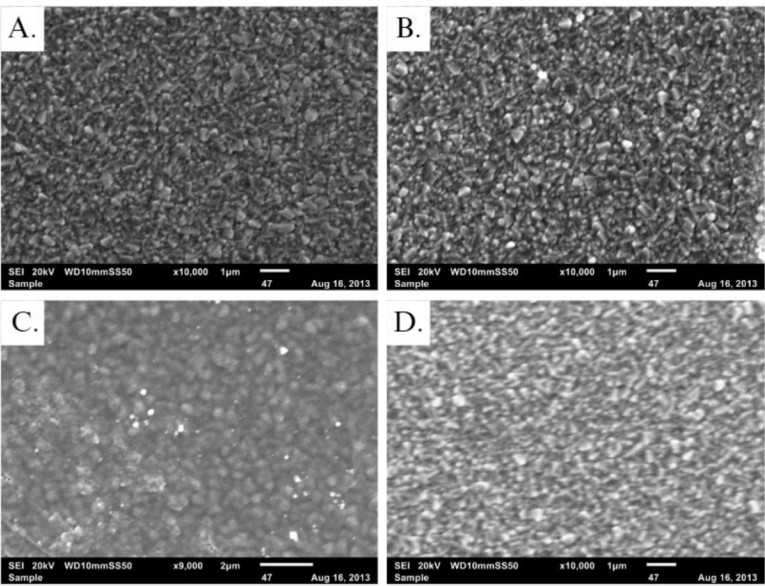

**Fig 2. SEM micrographs of (A) unmodified FTO-glass, (B) AgNPTs/FTO, (C) AgNPTs/Nf/FTO, and (D) washed FTO-glass.**

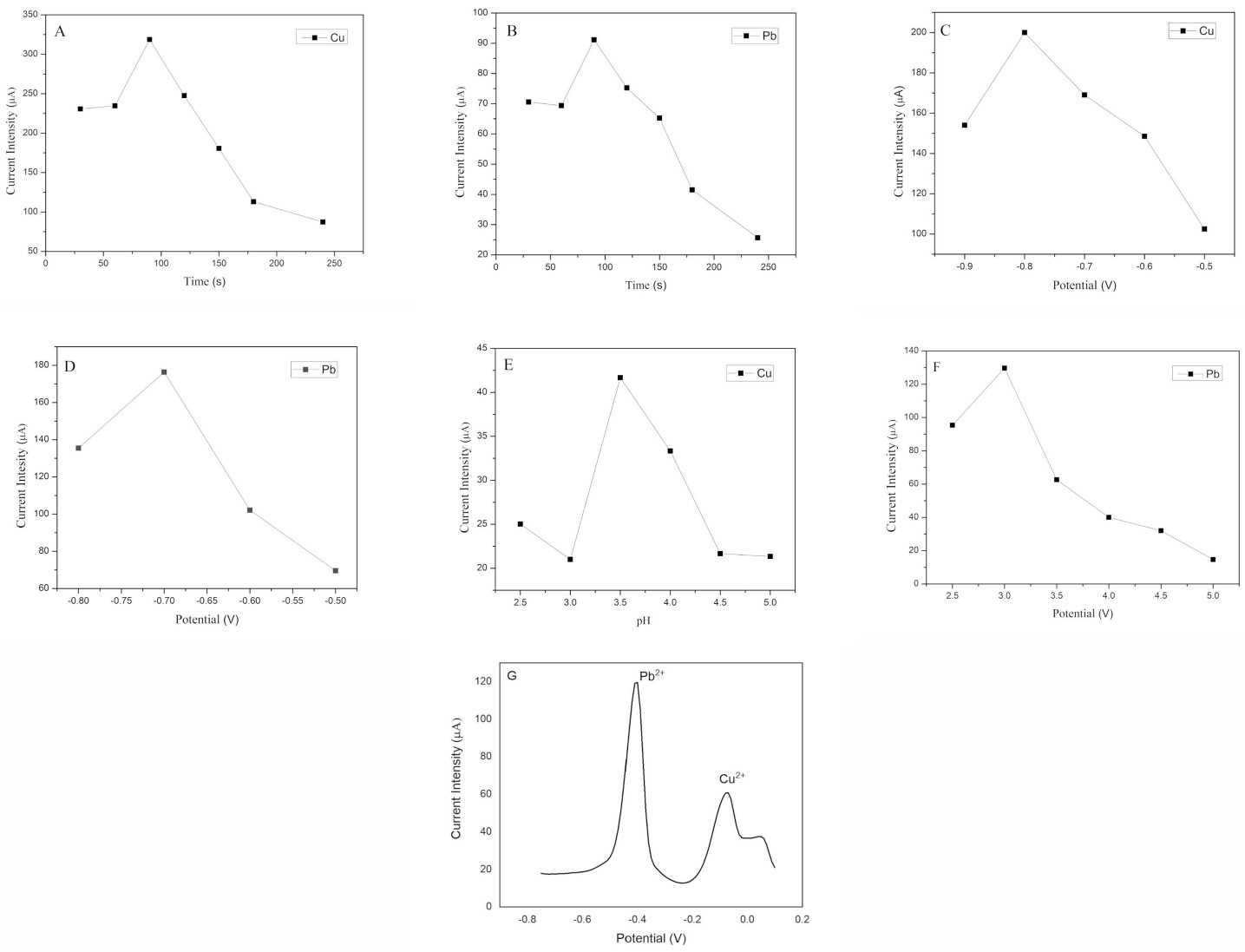

**Fig 3. Effect of the deposition time for (A) Cu (II) and (B) Pb (II), effect of potential applied for (C) Cu (II) and (D) Pb (II), effect of pH for (E) Cu (II) and (F) Pb (II), finally, (G) DPV of Cu (II) and Pb (II) at 60 ppb.**

pHs, reaching a maximum current at pH 3.5 for the Cu (II) ion and at pH 3 for the Pb (II) ion. After these points, a decrease in the signal of both metals was observed when using buffer solutions with more acidic pHs.

## Determination of Pb(II) and Cu(II) Ions

For the determination of Pb (II) and Cu (II) ions (Fig 3G), the results obtained in the optimization of this process were used, for which the measurements used the parameters of -0.7 V deposition voltage with 90 s of accumulation time in a PBS buffer solution at pH 3.5. With these optimized parameters, DPV analysis of stock solution of Pb (II) and Cu (II) was performed using working electrode at each stage of modification (unmodified FTO glass, Nf/FTO, and AgNPs/Nf/FTO) to confirm an improvement in sensor performance following the modifications. The results are shown in Fig 4.

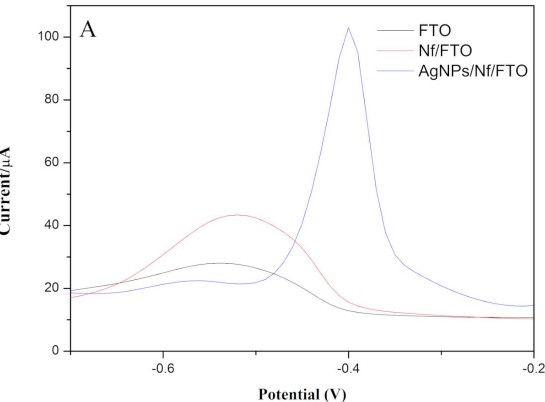
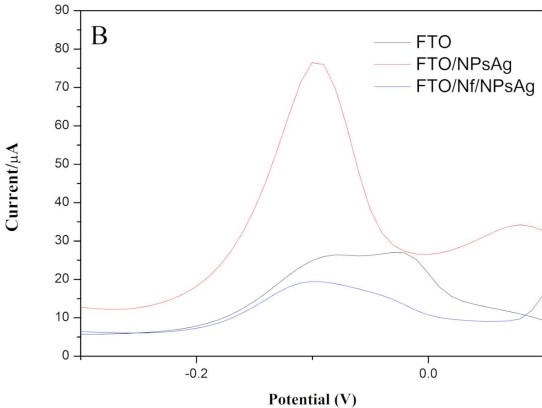

**Fig 4. Determination of (A) Pb (II) and (B) Cu (II) by Differential Pulse Voltammetry (DPV) using an unmodified FTO-glass, Nf/FTO, and AgNPs/Nf/FTO.**

For the simultaneous determination of both metal ions, stock solutions for both analytes were made with concentrations in the range of 10–60 ppb. Each of the measurements was performed in triplicate, obtaining the calibration curves shown in Fig 5A and 5B. For both curves, the limit of detection (LOD) was determined using the following formula [46]:

$$LOD = \frac{3.3SD}{S}$$

Where SD is the standard deviation of the values and S is the slope of the line.
The limit of quantification (LOQ) values were obtained using the following formula [46]:

$$LOQ = \frac{10SD}{S}$$

The LOD values for Cu and Pb ions were 3.41 and 9.80 ppb, respectively, while their LOQ values were 10.36 and 29.71 ppb, respectively.

As shown in Fig 5, for Cu (II) and Pb (II) the resulting calibration graphs are linear in the range 10–60 μg/L, for both metals, with corresponding equations of $y = 0.9267x – 4.2667$ and $y = 1.6565x + 5.1077$ and their determination coefficients, $R^2$, 0.9961 and 0.9882, respectively (x: concentration of analyte (μg/L) and y: current (μA).

## Determination of analytes in real samples

As an exploratory assessment of the sensor's performance, we conducted a preliminary comparison of the DPV technique with AgNPs/Nf/FTO-modified electrodes against the standard ICP-MS analytical technique for Cu (II) and Pb (II) determination in acid mine drainage samples. The results, summarized in Table 1, show a general agreement between DPV and ICP-MS methods for the concentrations tested. Specifically, for sample 1, DPV detected Cu (II) and Pb (II) ion concentrations of 1415 ± 3.51 μg/L and 78 ± 9.19 μg/L, respectively (Fig 6). For sample 2, DPV determined a concentration of 3778 ± 2.65 μg/L for Cu (II), while Pb (II) fell below the sensor's limit of quantification (LOQ). These results are consistent with those reported by SGS using ICP-MS.

While these initial findings suggest the sensor's viability in real-sample applications, they are not intended as a comprehensive comparative analysis. Due to the limited concentration

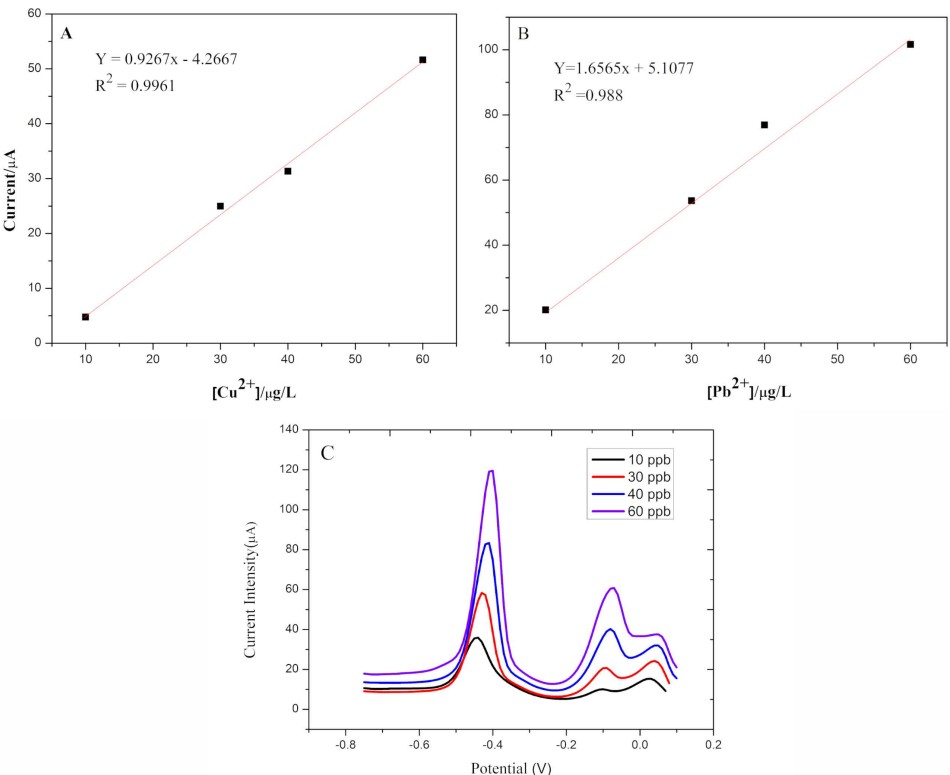

**Fig 5.** **(A) Calibration curve for Cu, (B) calibration curve for Pb, (C) simultaneous determination of Cu (II)and Pb (II) ions at concentrations from 10 ppb to 60 ppb using DPV on the AgNPs/Nf/FTO electrode.**

**Table 1. Comparison of DPV and ICP-MS estimated concentrations for Cu (II) and Pb (II) detected in acid mine drainage samples.**

| Detected analyte | Sample # | DPV concentration (µg/L) ± SD | ICP-MS concentration (µg/L) ± SD |
|---|---|---|---|
| Cu(II) | Sample 1 | 1415 ± 3.51 | 1528.3 ± 0.38 |
| | Sample 2 | 3778 ± 2.65 | 4079.52 ± 1.02 |
| Pb(II) | Sample 1 | 78 ± 9.19 | 83 ± 7.5 |
| | Sample 2 | <LOD | 2 ± 0.2 |

range and data points, the present comparison should be considered exploratory, providing insights into the potential of a AgNPs/Nf/FTO electrode for detecting Cu (II) and Pb (II) in complex matrices.

## Discussion

The electrochemical and structural characterization of FTO electrodes modified with silver nanoparticles and Nafion (AgNPs/Nf/FTO) unveiled crucial data on the effectiveness of our methodology in enhancing the detection of heavy metals. The observed decrease in current intensity in Nf/FTO electrodes, compared to pure FTO electrodes, underscores the obstructive effect of Nafion on electron transfer. Although this result could initially be interpreted as a disadvantage, it is important to consider that Nafion improves the adhesion of silver nanoparticles onto the electrode, which is essential for increasing the sensor's sensitivity. This duality

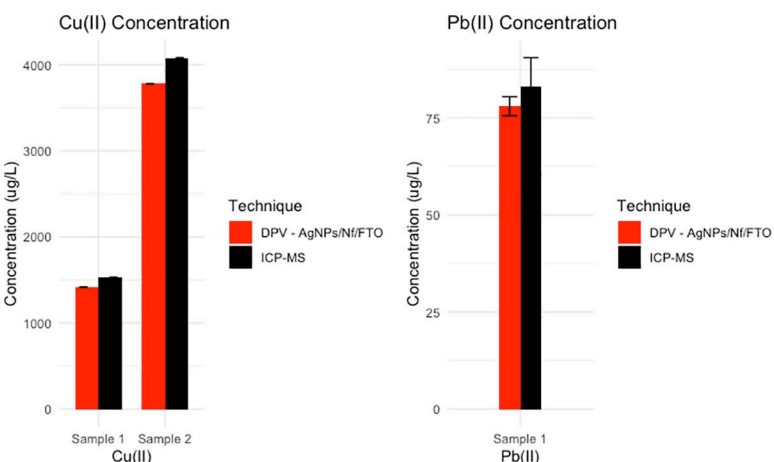

**Fig 6. Comparison between concentrations of Cu (II) and Pb (II) obtained from DPV and ICP-MS technique.**

reflects the complexity of designing electrode surfaces that balance conductivity and chemical functionality for the detection of specific analytes.

The novelty of utilizing FTO electrodes combined with Nafion and silver nanoparticles represents a significant advancement in the field of electrochemical sensors for heavy metal detection. This innovative approach leverages the unique properties of each component to create a synergistic effect, resulting in enhanced electrochemical performance. FTO provides a stable and conductive substrate, Nafion serves as an effective binding agent that also excludes interfering species, and the silver nanoparticles offer exceptional electrocatalytic activity. This combination has not only shown promising results in terms of sensitivity but also introduces a novel strategy for the development of high-performance electrochemical sensors.

Importantly, a sensor with these characteristics inherently possesses the capability to detect multiple metals, provided that their REDOX potentials are sufficiently differentiated to be discriminated by the potentiostat's resolution [35,47]. In our study, characteristic signals of Pb(II) at approximately -0.4 V and Cu(II) at -0.1 V are observed (Fig 3G), using an Ag/AgCl reference electrode. This adaptability suggests that such a tool has the potential to be utilized in a portable manner for various environmental monitoring applications, especially in remote Andean regions where heavy metal contamination is most problematic due to geological origins. The portability and versatility of this sensor could significantly enhance on-site detection and monitoring capabilities in areas that are currently underserved by traditional laboratory analysis methods.

The incorporation of AgNPs significantly enhances electron transfer, as indicated by the increased current intensity and the emergence of pronounced peaks in the voltammograms (Fig 1). This enhancement can be attributed to the high surface area of AgNPs, which facilitates redox processes, corroborating previous studies that reported significant improvements in the sensitivity of sensors based on metallic nanoparticles [36,39]. The results observed in Fig 4 present a comparison of the performance of the working electrode at each stage of its modification for the detection of Pb (II) (Fig 4A) and Cu (II) (Fig 4B). It is observed for both ions that the modification of the electrode with AgNPs enhances the signal in both cases for the same metal concentration in solution. These findings suggest that this approach surpasses the limitations of unmodified electrodes, particularly in terms of sensitivity and detection limits, suggesting a significant advantage over conventional methodologies [48].

Despite these promising results and potential applications, our system faces certain limitations. One major challenge is the potential for interference from other substances present in complex environmental matrices, which could affect the sensor's specificity and accuracy. However, the results in Table S1 show the presence of six additional metals detected in the total metal quantification performed using the ICP-MS technique by SGS. Nevertheless, despite the presence of these metals, there is no significant deviation between the reference value provided by the ICP-MS technique and the values reported by our sensor, which demonstrates a high level of selectivity. Furthermore, while the sensor demonstrates excellent sensitivity and selectivity for Pb (II) and Cu (II), distinguishing closely related metal ions with similar REDOX potentials may require further optimization of the electrode's surface properties or the use of advanced signal processing techniques. Additionally, the long-term stability and reproducibility of the sensor under real-world conditions remain critical areas for ongoing research.

To further underscore the practical implications of our findings, it is pertinent to mention that optimized sensors could have sufficiently high sensitivity and selectivity, which could be used for studies with environmental samples contaminated with heavy metals that could have concentrations reaching parts per billion (ppb), thus meeting the recommendations of the Environmental Quality Standards (ECA) given by regulatory agencies, as for example the Ministry of the Environment (MINAM) from Peru [49].

Furthermore, our research identifies the FTO-Nafion-silver nanoparticle combination as a promising and novel electrode configuration for electrochemical sensing applications. This insight not only contributes to the field by demonstrating the effectiveness of this specific composite material but also paves the way for future studies to explore and optimize the integration of these materials for broader analytical applications. The observed outstanding performance in heavy metal detection underscores the potential of this electrode design to serve as a benchmark for future sensor development.

The possibility of reusing electrodes after a simple washing protocol not only demonstrates the robustness of the electrode design but also its sustainability, a crucial aspect in the development of environmentally friendly detection technologies. This finding is consistent with the growing demand for greener and more economical analytical techniques in the field of electrochemistry.

The optimization of analytical parameters revealed the critical importance of accumulation time, deposition potential, and pH in maximizing the current signal. The selection of an optimal accumulation time of 90 seconds aligns with the theory that current signal increases with accumulation time due to the increase of ions on the electrode surface [35], however, very long times can lead to saturation of the electrode surface, a phenomenon observed in other metal preconcentration studies [32,35]. The choice of a deposition potential of -0.7 V reflects a balance between deposition efficiency and chemical stability, avoiding the reduction of tin oxide on the electrode [35].

Finally, to obtain the best signal for both metals, an optimal pH of 3.5 was selected. The decrease in the current signal for Pb (II) at pH 3 can be explained by the increased solubility of the metal complex [42]. Whereas the reduction of the signal at higher pHs reflects the formation of metallic hydroxide complexes resulting in a decrease of current [43].

The calibration curve (Fig 5) results of the present sensor show determination coefficient ($R^2$) values of 0.9961 and 0.9882 for Cu (II) and Pb (II) ions, respectively, for a linear range of 10–60 ppb. Furthermore, LOD values of 3.42 and 9.80 for such ions, respectively, were obtained, which showcases the sensor's capability to detect low analyte concentrations in water samples.

The results given by SGS in the determination of heavy metals (Table 1) in the water samples show an acceptable percentage error compared to the results of the ICP-MS technique.

For sample 1, 7.41% and 6.02% were obtained for the Cu and Pb ions, respectively, while in sample 2 a percentage error of 7.39% was obtained for the Cu ion, which revealed that the proposed sensor has potential for practical application.

The sensor's ability to simultaneously determine Pb (II) and Cu (II) with detection limits in the ppb range highlights its applicability for environmental monitoring, where the concentrations of these metals can be extremely low. These results not only comply with the ECA but also suggest that the sensor could be applicable for the detection of other heavy metals, offering a versatile and efficient method for assessing environmental pollution.

Compared to traditional techniques such as AAS and ICP-MS, our sensor offers a low-cost, rapid, and potential for in-situ analysis alternative, effectively addressing the limitations of cost and complexity of conventional methodologies. Additionally, the improved specificity and the ability to differentiate between oxidation states of the analyte promise superior selectivity, crucial for the precise determination of metals in complex environmental matrices.

In conclusion, this study demonstrates the potential of FTO electrodes modified with AgNPs and Nafion in environmental electrochemistry, also laying the groundwork for future research aimed at exploring the applicability of these sensors in analyzing a broader range of pollutants. This approach not only expands the spectrum of applicability of these sensors but also suggests exploring new material combinations to effectively address contemporary environmental challenges.

## Supporting information

**S1 Table. Analysis Inductively coupled plasma mass spectrometry (ICP-MS).** The ICP-MS analysis was performed by SGS del Perú S.A.C. The results were reported in the test report MA2411825 Rev. 0, which included the analysis of 49 metals, including Cu and Pb, from four acid mine drainage samples, using the EPA Method 200.8. The analyzed samples were labeled as M3-2 (Sample 1), and M4-1 (Sample 2). The results for samples 1 and 2 are presented, where copper and lead levels were detected in Sample 1, and only copper in Sample 2, using the AgNPs/Nf/FTO electrode
(PDF).

## Acknowledgments

We gratefully acknowledge Dr. Raúl Loayza and the CIMMA organization for their assistance in obtaining acid mine drainage samples for our sensor testing.

## Author contributions

**Conceptualization:** Leonardo J. Monroy-Cruz, Akemi Morales-Kato, Wilner Valenzuela, Mirko Zimic.

**Formal analysis:** Leonardo J. Monroy-Cruz, Akemi Morales-Kato, Yndira Dolores-Maldonado, Katiuska Castro, Alen Zimic-Sheen, María Belén Balta, Geraldine J. Otayza-Melgarejo, Raúl León, Wilner Valenzuela.

**Funding acquisition:** Patricia Sheen, Mirko Zimic.

**Investigation:** Leonardo J. Monroy-Cruz, Akemi Morales-Kato, Yndira Dolores-Maldonado, Katiuska Castro, Alen Zimic-Sheen, María Belén Balta, Geraldine J. Otayza-Melgarejo, Wilner Valenzuela, Mirko Zimic.

**Methodology:** Leonardo J. Monroy-Cruz, Akemi Morales-Kato, Yndira Dolores-Maldonado, Katiuska Castro, Geraldine J. Otayza-Melgarejo, Raúl León, Wilner Valenzuela, Mirko Zimic.

**Project administration:** Patricia Sheen, Mirko Zimic.

**Resources:** Raúl León, Patricia Sheen, Mirko Zimic.

**Supervision:** Patricia Sheen, Wilner Valenzuela, Mirko Zimic.

**Validation:** Leonardo J. Monroy-Cruz, Akemi Morales-Kato, Yndira Dolores-Maldonado, Katiuska Castro, Alen Zimic-Sheen, María Belén Balta, Geraldine J. Otayza-Melgarejo, Raúl León, Wilner Valenzuela.

**Visualization:** Leonardo J. Monroy-Cruz, Akemi Morales-Kato, Yndira Dolores-Maldonado, Katiuska Castro, Alen Zimic-Sheen, María Belén Balta, Raúl León, Wilner Valenzuela, Mirko Zimic.

**Writing – original draft:** Leonardo J. Monroy-Cruz, Akemi Morales-Kato, Yndira Dolores-Maldonado, Katiuska Castro, Alen Zimic-Sheen, María Belén Balta, Geraldine J. Otayza-Melgarejo, Patricia Sheen, Wilner Valenzuela, Mirko Zimic.

**Writing – review & editing:** Leonardo J. Monroy-Cruz, Akemi Morales-Kato, Yndira Dolores-Maldonado, Katiuska Castro, Alen Zimic-Sheen, María Belén Balta, Patricia Sheen, Wilner Valenzuela, Mirko Zimic.

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
