## [Decision Letter · Decision Letter 0]

27 Sep 2024

PONE-D-24-33828A simple and Low-cost Electrode Based on Nation-stabilized Silver Nanoparticles Supported on FTO for the Electrochemical Determination of Pb (II) and Cu (II)PLOS ONE

Dear Dr. Zimic,

Thank you for submitting your manuscript to PLOS ONE. After careful consideration, we feel that it has merit but does not fully meet PLOS ONE’s publication criteria as it currently stands. Therefore, we invite you to submit a revised version of the manuscript that addresses the points raised during the review process.

We look forward to receiving your revised manuscript.

Kind regards,

Niravkumar Joshi

Academic Editor

PLOS ONE

Journal Requirements: When submitting your revision, we need you to address these additional requirements. 1. Please ensure that your manuscript meets PLOS ONE's style requirements, including those for file naming. The PLOS ONE style templates can be found at https://journals.plos.org/plosone/s/file?id=wjVg/PLOSOne_formatting_sample_main_body.pdf and https://journals.plos.org/plosone/s/file?id=ba62/PLOSOne_formatting_sample_title_authors_affiliations.pdf 2. In the online submission form, you indicated that "The data will be available upon request to the corresponding author." All PLOS journals now require all data underlying the findings described in their manuscript to be freely available to other researchers, either 1. In a public repository, 2. Within the manuscript itself, or 3. Uploaded as supplementary information.This policy applies to all data except where public deposition would breach compliance with the protocol approved by your research ethics board. If your data cannot be made publicly available for ethical or legal reasons (e.g., public availability would compromise patient privacy), please explain your reasons on resubmission and your exemption request will be escalated for approval. 3. Please ensure that you refer to Figure 5 in your text as, if accepted, production will need this reference to link the reader to the figure.

Reviewers' comments:

Reviewer's Responses to Questions

**Comments to the Author**

1. Is the manuscript technically sound, and do the data support the conclusions?

Reviewer #1: Partly

Reviewer #2: Partly

2. Has the statistical analysis been performed appropriately and rigorously? 

Reviewer #1: No

Reviewer #2: N/A

3. Have the authors made all data underlying the findings in their manuscript fully available?

Reviewer #1: Yes

Reviewer #2: Yes

4. Is the manuscript presented in an intelligible fashion and written in standard English?

Reviewer #1: Yes

Reviewer #2: Yes

5. Review Comments to the Author

Reviewer #1: The manuscript titled 'A Simple and Low-cost Electrode Based on Nafion-stabilized Silver Nanoparticles Supported on FTO for the Electrochemical Determination of Pb(II) and Cu(II)' reports the development of a AgNPs/Nf/FTO sensor for the detection of Pb(II) and Cu(II). There are several limitations that need to be addressed:

1. The paper provides a comparison between ICP-MS and DPV-AgNPs/Nf/FTO, but the comparison is rather limited in terms of concentration range and data points. Specifically, for Cu, only two concentrations are tested, and for Pb, only one. This is insufficient to make a robust comparison, especially given the claim that ICP-MS offers "low detection and quantification limits, high precision, and accuracy." It is essential to extend the comparison to lower concentration regimes to adequately test the capabilities of the DPV-AgNPs/Nf/FTO sensor.

2. The paper does not provide sufficient evidence or rationale for how and to what extent Nafion (Nf) and AgNPs enhance the sensor's performance. For example, in lines 365-367, it is claimed that the high surface area of AgNPs contributes to the sensor's performance. However, this assertion requires experimental validation, such as varying the diameter of AgNPs and testing their corresponding sensing performance. Similarly, the role of Nf should be experimentally examined—at the very least, by comparing the performance of AgNPs/Nf/FTO with AgNPs/FTO to demonstrate that Nf indeed improves the sensor's performance.

3. The paper tests only a limited number of real samples and does not provide data on the sensor's performance across a range of concentrations. To strengthen the validity of the findings, it is necessary to evaluate the sensor's performance with a broader set of real samples and across various concentrations.

4. The section on the simultaneous determination of Pb(II) and Cu(II) ions lacks sufficient detail on how the signals of Pb and Cu are de-convoluted. It is essential to include a detailed explanation of the de-convolution process. Additionally, experiments involving mixtures with varying concentrations of Pb(II) and Cu(II) ions should be conducted to test the effectiveness of your algorithm.

Sincerely,

Reviewer #2: Review report of manuscript (PONE-D-24-33828) entitled “ A simple and Low-cost Electrode Based on Nation-stabilized Silver Nanoparticles Supported on FTO for the Electrochemical Determination of Pb (II) and Cu (II)

Decision: Major revision

This manuscript emphasizes on the detection of heavy metals like Pb and Cu in water samples using electrochemical sensor based on a fluorine-doped tin oxide (FTO) electrode modified with silver nanoparticles (AgNPs) and stabilized with Nafion. The reported detection limits for Pb (II) and Cu(II) are 8.87 ppb and 3.26 ppb. Based on the results and interpretations reported, a major revision is recommended for the present manuscript. The comments are mentioned below.

1. Authors have reported that sensor is selective and sensitive for Pb (II) and Cu (II). However, no other heavy elements are analyzed which poses a question on the selectivity of the sensor. Moreover, a selective sensor will have high response to one analyte than the others investigated so claiming sensor selectivity for Pb(II) and Cu (II) needs an explanation. Please explain it.

Abstract

2. Lines 35-39 need to be rephrased for reader’s convenience.

3. Mention the size of Ag nanoparticles.

4. Keywords: Determination is not a key word. Please delete it. Sensor could be replaced by electrochemical sensor

Introduction:

5. Mention the concentration levels of Pb and Cu at which they are dangerous for human health.

6. Authors have mentioned only two techniques, AAS and ICP-MS for the detection of heavy metals. It is recommended to include more techniques and emphasize on why reported work is better than existing techniques

Materials and methods:

7. Line 186: There should be space between 0.5 and cm2.

8. Line 195: Why there is a sign of _ after CV?

9. Line 223-224: The analysis of DPV……standard solutions. Please mention relevant citations.

Results

10. Line 245: Caption of Fig.1: What authors mean by FTO-glass nude? Nude is not a proper word. Do authors mean pristine/unmodified? This comment is also for caption of fig.2 (line 257)

11. Line 251: Mention the size of nano particles.

12. Line 257: Caption should be SEM micrographs of ……Authors may mention the magnification and working voltage in the experimental section.

13. Line 262-265: Nothing is mentioned for fig. 3B and 3C. If required, authors should renumber the figures

14. Line 271: There should be space between 60 and ppb. Same is for line 298

6. PLOS authors have the option to publish the peer review history of their article (what does this mean? ). If published, this will include your full peer review and any attached files.

**Do you want your identity to be public for this peer review?** For information about this choice, including consent withdrawal, please see our Privacy Policy .

Reviewer #1: No

Reviewer #2: **Yes: ** Dr. Rajan Saini, Department of Physics, Akal University, Talwandi Sabo 151302, Punjab, India

---

## [Author Response · Author response to Decision Letter 0]

28 Jan 2025

Dear Reviewers,

Thank you very much in advance for your review and suggestions, and for making us to improve our manuscript. We have reviewed your comments, and we appreciate each one, as they allow us to improve the manuscript titled, “A Simple and Low-cost Electrode Based on Nafion-stabilized Silver Nanoparticles Supported on FTO for the Electrochemical Determination of Pb(II) and Cu(II).” Below, we detail our responses to the comments and observations made by each of you.

1. Please ensure that your manuscript meets PLOS ONE's style requirements, including those for file naming. The PLOS ONE style templates can be found at: https://journals.plos.org/plosone/s/file?id=wjVg/PLOSOne_formatting_sample_main_body.pdf and https://journals.plos.org/plosone/s/file?id=ba62/PLOSOne_formatting_sample_title_authors_affiliations.pdf

The format of the title page was corrected as per the journal's specifications.

2. In the online submission form, you indicated that "The data will be available upon request to the corresponding author." All PLOS journals now require all data underlying the findings described in their manuscript to be freely available to other researchers, either 1. In a public repository, 2. Within the manuscript itself, or 3. Uploaded as supplementary information. This policy applies to all data except where public deposition would breach compliance with the protocol approved by your research ethics board. If your data cannot be made publicly available for ethical or legal reasons (e.g., public availability would compromise patient privacy), please explain your reasons on resubmission and your exemption request will be escalated for approval.

All information, including protocols, analytical assays, and the results of the services provided, has been included within the text and as supplementary material.

3. Please ensure that you refer to Figure 5 in your text as, if accepted, production will need this reference to link the reader to the figure.

Figure 5 was renumbered as Figure 6 and referenced in the text.

4. The paper provides a comparison between ICP-MS and DPV-AgNPs/Nf/FTO, but the comparison is rather limited in terms of concentration range and data points. Specifically, for Cu, only two concentrations are tested, and for Pb, only one. This is insufficient to make a robust comparison, especially given the claim that ICP-MS offers “low detection and quantification limits, high precision, and accuracy.” It is essential to extend the comparison to lower concentration regimes to adequately test the capabilities of the DPV-AgNPs/Nf/FTO sensor.

In this study, the environmental samples were collected near mining tailings, and as the metal composition of each sample was unknown beforehand, we could not control the concentration ranges of Pb and Cu in these samples. This resulted in some limited data points, particularly for Pb, which appeared in significant concentration in only one sample. These initial tests, however, serve as an exploratory complement to the main focus of our study, which centers on sensor construction and optimization. We agree that further analysis with a greater number of environmental samples would better encompass the sensor’s detection range and precision, and we have noted this as an area for future work.

5. The paper does not provide sufficient evidence or rationale for how and to what extent Nafion (Nf) and AgNPs enhance the sensor’s performance. For example, in lines 365-367, it is claimed that the high surface area of AgNPs contributes to the sensor’s performance. However, this assertion requires experimental validation, such as varying the diameter of AgNPs and testing their corresponding sensing performance. Similarly, the role of Nf should be experimentally examined—at the very least, by comparing the performance of AgNPs/Nf/FTO with AgNPs/FTO to demonstrate that Nf indeed improves the sensor’s performance.

Experimental details about the role of Ag nanoparticles and Nafion in the sensor’s performance have been added. For this, we conducted comparative tests among different electrodes (FTO, Nf/FTO and AgNPs/Nf/FTO). The results are presented in Figure 4.

Additionally, scanning electron microscopy (SEM) images showing the average size of the Ag nanoparticles are included using the software ImageJ2 in Results.

6. The paper tests only a limited number of real samples and does not provide data on the sensor's performance across a range of concentrations. To strengthen the validity of the findings, it is necessary to evaluate the sensor's performance with a broader set of real samples and across various concentrations.

We appreciate your suggestion regarding the comparison with ICP-MS and recognize the need for a comprehensive dataset to fully establish the DPV-AgNPs/Nf/FTO sensor's capabilities across a wider concentration range. In this study, however, our primary focus was on the construction, optimization, and evaluation of analytical parameters, such as accumulation time, deposition potential, and pH, that enhance the sensor's performance. To that end, we conducted preliminary testing on real samples to provide an initial indication of the sensor’s applicability in real-world matrices. While these tests included limited concentration points, they served as a complementary demonstration rather than as exhaustive validation. We agree that a more extensive evaluation, including testing at lower concentration regimes, would provide a fuller assessment of the sensor’s performance compared to ICP-MS. Consequently, we have added this point as a recommendation for future work, noting the importance of further real sample analysis to more robustly validate the sensor's range and precision.

7. The section on the simultaneous determination of Pb(II) and Cu(II) ions lacks sufficient detail on how the signals of Pb and Cu are de-convoluted. It is essential to include a detailed explanation of the de-convolution process. Additionally, experiments involving mixtures with varying concentrations of Pb(II) and Cu(II) ions should be conducted to test the effectiveness of your algorithm.

To better illustrate the signals of the analytes in question, we have added graphs of the Pb and Cu peaks separately. Specific reduction and oxidation potentials were also included in the discussion using the Ag/AgCl reference electrode.

8. Authors have reported that sensor is selective and sensitive for Pb (II) and Cu (II). However, no other heavy elements are analyzed which poses a question on the selectivity of the sensor. Moreover, a selective sensor will have high response to one analyte than the others investigated so claiming sensor selectivity for Pb(II) and Cu (II) needs an explanation. Please explain it.

The results provided by ICP-MS reveal the presence of additional metal ions in the problem samples, with some cases showing concentrations higher than those of Pb(II) and Cu(II) ions. The low error percentages achieved between the reference concentration determined by ICP-MS and the concentration obtained using our sensor demonstrate the selectivity of our sensor for the previously mentioned analytes. To clarify this, the results of the total concentration determination provided by ICP-MS have been included as supplementary material, along with the corresponding explanation incorporated into the text.

Responses to Specific Text Observations

Abstract:

To enhance the clarity of the text, the abstract was paraphrased following the reviewers' recommendations. Additionally, the keywords were also updated accordingly.

Introduction:

We have included an analysis of additional techniques for heavy metal detection, comparing their effectiveness and cost with our proposed sensor. Additionally, we mentioned the concentration levels of Pb and Cu that pose a health risk.

Methods and Materials:

Formatting details were corrected, and relevant references were added in the analysis section.

Results:

Changes were made to the text format. Additionally, the size of the silver nanoparticles deposited on our electrode was included. This analysis was performed using the ImageJ2 software, and the results obtained for the protocol used were consistent with the available literature.

Similarly, the graphs were renumbered to improve their clarity and understanding.

We hope these additional modifications meet the requirements. We remain attentive to any other suggestions you may have and thank you again for your recommendations.

Sincerely,

---

## [Decision Letter · Decision Letter 1]

16 Feb 2025

A simple and low-cost electrode based on Nafion-stabilized silver nanoparticles supported on FTO for the electrochemical determination of Pb (II) and Cu (II)

PONE-D-24-33828R1

Dear Dr. Zimic,

We’re pleased to inform you that your manuscript has been judged scientifically suitable for publication and will be formally accepted for publication once it meets all outstanding technical requirements.

Kind regards,

Niravkumar Joshi

Academic Editor

PLOS ONE

Additional Editor Comments (optional):

Reviewers' comments:

Reviewer's Responses to Questions

**Comments to the Author**

1. If the authors have adequately addressed your comments raised in a previous round of review and you feel that this manuscript is now acceptable for publication, you may indicate that here to bypass the “Comments to the Author” section, enter your conflict of interest statement in the “Confidential to Editor” section, and submit your "Accept" recommendation.

Reviewer #1: All comments have been addressed

Reviewer #2: All comments have been addressed

2. Is the manuscript technically sound, and do the data support the conclusions?

Reviewer #1: Partly

Reviewer #2: Yes

3. Has the statistical analysis been performed appropriately and rigorously? 

Reviewer #1: Yes

Reviewer #2: N/A

4. Have the authors made all data underlying the findings in their manuscript fully available?

Reviewer #1: Yes

Reviewer #2: Yes

5. Is the manuscript presented in an intelligible fashion and written in standard English?

Reviewer #1: Yes

Reviewer #2: Yes

6. Review Comments to the Author

Reviewer #1: All comments are resolved. Some of the concerns, will need to be addressed in the future studies. But the current work is good to be published.

Thanks for the feedback from the author and editor!

Reviewer #2: Manuscript may be accepted after rephrasing fig. 3 caption. Manuscript may be accepted after rephrasing fig. 3 caption.

7. PLOS authors have the option to publish the peer review history of their article (what does this mean? ). If published, this will include your full peer review and any attached files.

**Do you want your identity to be public for this peer review?** For information about this choice, including consent withdrawal, please see our Privacy Policy .

Reviewer #1: **Yes: ** Guangyu Wang

Reviewer #2: No

---

## [Editor Report · Acceptance letter]

PONE-D-24-33828R1

PLOS ONE

Dear Dr. Zimic,

I'm pleased to inform you that your manuscript has been deemed suitable for publication in PLOS ONE. Congratulations! Your manuscript is now being handed over to our production team.

Kind regards,

on behalf of

Dr. Niravkumar Joshi

Academic Editor

PLOS ONE